# Evaluation of the Quality and Lipid Content of Artisan Sausages Produced in Tungurahua, Ecuador

**DOI:** 10.3390/foods12234288

**Published:** 2023-11-28

**Authors:** Lander Pérez, Rosa Pincay, Diego Salazar, Nelly Flores, Consuelo Escolastico

**Affiliations:** 1International School of Doctorate, Sciences Doctorate, Universidad Nacional de Educación a Distancia (UNED), E-28040 Madrid, Spain; cescolastico@ccia.uned.es; 2G+ Biofood and Engineering Research Group, Food and Biotechnology Faculty, Technical University of Ambato (UTA), Av. Los Chasquis y Río Payamino, Ambato 180206, Ecuador; rpincay5588@uta.edu.ec (R.P.); dm.salazar@uta.edu.ec (D.S.); 3Research and Development Directorate, Food and Biotechnology Faculty, Technical University of Ambato (UTA), Av. Los Chasquis y Río Payamino, Ambato 180206, Ecuador; ne.flores@uta.edu.ec

**Keywords:** artisan sausages, fatty acids, nutritional properties, sensorial properties

## Abstract

The consumption of sausage worldwide increases every year; because of this increase, artisanal products have appeared and are intended to be perceived as natural and healthy. Obesity and cardiovascular diseases associated with consuming meat and meat derivatives have been estimated to be the leading cause of death in several countries. This study aimed to evaluate the nutritional quality, lipid content, and presence of saturated and unsaturated fatty acids, contributing to demonstrating the real nutritional value of artisanal sausages produced in Ecuador. Sausages from 10 factories in Ambato, Pelileo, and Píllaro, located in Tungurahua, Ecuador, were evaluated. The pH and acidity, color, proximal, sensory, microbiological, and lipid content were assessed. The pH and acidity showed a slight variation in all of the samples. Proximal analysis (moisture, protein, fat, and ash) established that the artisan sausages did not differ from the type of sausages reported in the literature. Microbiological analyses showed a good microbial quality, and there was no presence of *Staphylococcus aureus*, *Enterobacteria*, molds, or yeasts. The sensory attributes were similar for all of the sausages; the panelists did not notice any strange taste or odor. The lipid content showed that the artisanal sausages contained the highest percentage of palmitic, stearic, elaidic, and linolelaidic fatty acids. Unsaturated fatty acids were the most prevalent in all of the sausages collected from different locations. The results showed that the nutritional, microbiological, and sensory quality of the artisanal sausages did not show any parameter that would allow them to be classified as different or as having a better nutritional value.

## 1. Introduction

Currently, there is a growing emphasis on improving the nutritional balance of food. In this context, artisanal or organic products have gained popularity as they are perceived by the population as being more natural. However, it is important to note that not all these products are nutritious and healthy. Furthermore, these trends align with the understanding that attitudes related to environmental concerns and personal health significantly influence people’s willingness to pay for organic food [1]. Historically, some of the most consumed foods have included meat products such as Frankfurter, Bratwurst, and Vienna, among other sausages, which are losing market value due to deficient health components such as fiber, or as a result of high salt, nitrite, and fat contents [2,3,4]. Consumers perceive this type of food as unhealthy and related to diseases such as diabetes, heart disease, stroke, and some forms of cancer [5,6]. In this sense, researchers are studying new formulations to improve the nutritional quality of products and provide new formulations, ingredients, and nutritional additives to these producers.

Recent research has revealed that adjusting sausage formulations, such as through the inclusion of bamboo fibers to reduce the fat content in bologna sausages, is a viable alternative [7], as well as the enhancement of Frankfurter-type sausages enriched with buckwheat as a source of bioactive compounds [8] and a reduction or substitution in sodium additives in cooked sausages [9]. It is widely acknowledged that the development of novel meat formulations is prohibitively expensive, leading to hesitation among some manufacturers. In emerging markets, certain producers opt for a more straightforward approach by altering the product label to create the impression of a “new” offering, thereby attracting a fresh consumer base [10].

The quality of sausages depends on critical factors, notably their fat, nitrite, and salt contents. Salt, a standard component of the human diet, has been associated with health concerns, particularly hypertension and other related issues [2]. Quality sausages are determined by the content of fat, nitrites, and salt content. Salt is a food preservative and flavoring component; however, the excessive consumption of salt in the human diet is associated with hypertension and other health problems [2,11]. However, consumers are reluctant to compromise on the taste and flavor profiles of meat products; they are not ready to change the taste and flavor of meat products [12]. Additionally, consuming meat and meat products has been linked to increased risk of colorectal cancer, primarily attributed to nitrites [13]. Notably, the fat content poses a significant challenge in the reformulation of meat products because of its influence on flavor. Moreover, it is the most extensively studied component concerning its impact on public health [14]. In sausage production, fat plays an essential role in enhancing attributes such as juiciness, tenderness, and overall palatability [15]. Traditionally, the fat in sausages is sourced from pork back fat and intramuscular fat. However, the nutritional quality of these fat sources, characterized by their high fatty acid composition, has recently become a primary concern, categorizing them as detrimental to human health [16,17]. The abundance of free fatty acids, including oleate, palmitate, stearate, myristate, palmitoleate, and linoleate, is associated with hepatic lipotoxicity, predominantly due to the excessive consumption of saturated and trans-fatty acids [18].

Consumers, based on messages seeking to improve their health, have focused on searching for foods similar to those they have usually consumed, but that somehow have a better nutritional value and that can be consumed with a low risk of developing diseases. This consumer requirement has generated fraudulent practices by labeling a false idea of nutritional foods, which, of course, are not true. In recent years, the concept of food fraud scandals has been associated with mislabeling practices [19]. Although this dishonest practice is difficult to prove, many consumers are guided by the perception that a food labeled as nutritional, organic, or artisanal has a different nutritional value that will not affect their health or generate possible illnesses [20,21]. Thus, the objective of this work is to evaluate the quality of sausages labeled as artisanal produced in 10 factories in Tungurahua, Ecuador.

## 2. Materials and Methods

### 2.1. Materials

Sausages were procured from 10 distinct factories located across various cities in Ecuador. Specifically, Ambato was the source of sausages from six factories, Pelileo contributed sausages from three factories, and one factory in Píllaro provided the remaining samples. Each factory supplied 200 g of sausages, which were subsequently refrigerated and stored at temperatures ranging from 4 to 6 °C.

To gain insight into sausage production processes, comprehensive observations were conducted at these diverse facilities. A uniform procedure for sausage production was employed across all of the evaluated factories. The common manufacturing approach involved the following steps: First, meat consisting of beef and pork, as well as pork back fat, was mechanically minced. Subsequently, the minced meat and fat, along with the requisite seasonings and crushed ice, were homogenized within a cutter. The resulting mixture was carefully stuffed into an artificial casing. The sausages were subsequently cooked in a water bath, maintaining temperatures in the range of 80–85 °C until the core of the product reached 73 °C, a process that typically lasted approximately 30 min. Following the heating phase, the sausages were subjected to rapid cooling in a cold bath, with the objective of reaching an internal temperature of 30 °C. Subsequently, the sausages were packaged in high-density polyethylene plastic food bags and stored under refrigeration, while maintaining the temperature within the 4–6 °C range.

### 2.2. Quality Determination in Sausages

#### 2.2.1. Proximal Analysis

The proximate composition (moisture, ash, protein, and fat) was evaluated following the official methods AOAC 19 927.05, AOAC 923.03, AOAC 2001.11, and AOAC 2033.06, respectively (AOAC, 2005) [22]. The carbohydrate content was estimated using difference and expressed as g/100 g. The moisture content (%g/100 g) was determined by drying in a stove at 105 °C ± 2 °C, the nitrogen was established using the Kjeldahl method, and the protein content (g/100 g) was estimated with the nitrogen portion using a 6.25 factor. The fat content (g/100 g) was determined using the Soxhlet method and the ash percentage was determined through incineration in a muffle at 550 °C. The water activity was determined using the aw equip (AQUA LAB Dew Point Water Activity Meter 4TE); the circular container of the equipment was filled with ultra-pure Milli-Q (Thermo Fisher Scientific, Waltham, MA, USA) water for calibration, and it was leveled with the sample, previously ground with a pestle, and placed into the equipment for measuring. The titratable acidity was evaluated in a titrator (Mettler Toledo G20 Compact Titrator); 10 g of sample was mixed with 60 mL of ultrapure Milli-Q (Thermo Scientific) to obtain the sample to be evaluated, and the acidity was reported as the lactic acid content (g/100 g of lactic acid). Furthermore, a potentiometer measured the pH (Mettler Toledo Seven Compact). All of the determinations were performed in triplicate using three samples for each treatment.

#### 2.2.2. Energy Value

The calorie content was estimated using 100 g of sample, where the overall sum of calories of the individual components was the energy value for each component: fat (X 9 kcal/g), protein (X 4 kcal/g), and carbohydrate (X 4 kcal/g) [23,24].

#### 2.2.3. Color

Color parameters were measured using a Hunter Lab Colorimeter (mini Scan 4500 L EZ, Hunter Associates Laboratory INC, Reston, VA, USA) calibrated with an illuminator D65 (natural light) and standard observer D10. The colorimeter was calibrated before taking the measurements, and the white tile standard was used. The chroma polar coordinate or saturation C* was calculated using the expression C* = √ (a*^2^ + b*^2^) and hue tone using (h*) = arctang (b*/a*) to a* and b* positives. The chroma and hue tone were calculated based on Salazar, Arancibia, Calderón, López-Caballero, and Montero [1]. At least 15 measurements were performed in different areas of the sample, and the averages were recorded as the reported values.

#### 2.2.4. Microbiological Analysis

Here, 10 g of artisan sausages and 90 mL of buffered 0.1% peptone water (Difco, Le Pont de Claix, France) were placed into a sterile bag and homogenized in a stomacher homogenizer (Model 400 C, Seward, London, UK) for 1 min at room temperature. Serial decimal dilutions were prepared in peptone water for each sample. *Enterobacteria* were determined on a double layer of Violet Red Bile Glucose Agar (VRBG) (Acumedia, MI, USA) incubated at 30 °C/72 h. Mesophilic aerobic bacteria were determined in PCA agar (Difco) incubated at 30 °C for 72 h. *Staphylococcus aureus* were determined on Baird Parker agar (Difco) supplemented with egg yolk tellurite incubated at 30 °C for 48 h. Mold and yeast were spread plated on Rose Bengal Agar (RBC) (Difco) and incubated at 25 °C. All of the assays were performed according to the Bacteriological Analytical Manual of the U.S. Food and Drug Administration [25]. All of the analyses were performed in triplicate.

#### 2.2.5. Sensory Quality

Sensorial parameters were evaluated according to the procedure described by Salazar, Arancibia, Calderón, López-Caballero, and Montero [1]. Thirty trained panelists assessed attributes such as the overall acceptability, taste, and odor of the sausages. The panel received training in previous sessions. The sensory evaluation was developed in a sensorial cabin illuminated with white light, the temperature in the room was 20 °C, and the panelists used a test based on a 5-point hedonic scale (1—disliked very much; 2—disliked moderately; 3—neither liked nor disliked; 4—liked moderately; 5—liked very much). For the evaluation, two cylinders (1.6 cm in diameter _ and 3 cm in length) of grilled samples without casing were provided to the panelist for assessment. A glass of water and salted crackers for mouth cleaning were also provided.

#### 2.2.6. Chlorides Determination

The determination of chlorides was carried out according to the Mohr method [26]. Approximately 1 g of each sample was weighed into an Erlenmeyer flask, and an ultrapure Milli-Q water mixture (Thermo Scientific) was added and boiled to extract the sodium chloride. The chlorides were titrated with 0.1 N of silver nitrate according to the Mohr method, using two drops of 5% potassium chromate as an indicator. The analysis of each sample was developed in triplicate.

#### 2.2.7. Nitrite Determination

Nitrite determination was developed according to the AOAC 973.331 method. The measurement was based on spectrophotometric detection of a chromophore formed by the reaction between nitrites and 1-naphthyl-ethylene diamine (NED) sulfanilamide. Approximately 1 g of previously ground sample was weighed and diluted with ultra-pure Milli-Q (Thermo Scientific) water. The mixture was transferred to 50 mL volumetric flasks, and two washes were completed with 2 mL of hot water. The volumetric flasks with the samples were kept in a water bath for 2 h, then the aqueous extract was obtained at room temperature; finally, it was bottled and filtered. Then 2 mL of the filtered fluid was transferred into a 10 mL flask containing 0.5 mL sulfanilamide; after that, the mixture was shaken. After 5 min, 0.5 mL of Eriochrome Black T (EBT) chemical indicator was added, and the absorbance was measured at 540 nm in quartz cells in a UV−VIS (Hach, DR 5000) spectrometer. Three analyses of each sample were developed.

### 2.3. Lipid Content Quantification

#### 2.3.1. Fat Extraction

The method proposed by Bligh and Dyer [27] was used: 6.7 mL of chloroform, 1.7 mL of extra pure methanol (99.5%), 1.4 mL of ultra-pure Milli-Q water were added; all of the tubes were prepared into the fume hood; and the tubes were covered with aluminum foil and placed in a horizontal agitator for 1 h at 280 rpm and then centrifuged for 10 min at 5000× *g* rpm. The tubes were placed to rest for 72 h to facilitate the separation of the two phases. The organic phase was placed in the rotary evaporator (Eyela N-400 H, Shanghai, China) at 33 °C with 26% vortex velocity and 400 millibars of pressure to recover the solvent.

#### 2.3.2. Methyl Esterification of Fatty Acids

Methyl esterification was developed based on the method proposed by Ichihara and Fukubayashi [28]. A sample of the previously extracted fat was taken, and 2 mL of potassium hydroxide with methanol 0.5 M was added. The mixture was placed in a boiling water bath for 10 min. It was then allowed to cool at room temperature, 1 mL of methanolic hydrochloric acid (1: 4 *v*/*v*) was placed in it, and it was subsequently maintained in a water bath of 50 °C for 25 min. Finally, 3 mL of ultra-pure Milli-Q water and 10 mL of chromatographic grade hexane were added and homogenized.

#### 2.3.3. Chromatographic Determination

For the chromatography reading, the extract obtained previously was left to rest for a minimum of 2 h intervals and 1.5 mL of the hexane extract was obtained. Finally, the samples were filtered using microfilters (Econofilter) and placed into Agilent Technology vials for analysis [29,30]. The chromatographic conditions for quantifying the fatty acids were as follows: volume of injection of 0.3 µL and the temperature of the injector was 250 °C. Split injection at 15 mL/min. An Agilent HP-88 Column (Length: 60 m; diameter; 0.25 mm; thickness: 0.25µm). Flux of 1.4 mL/min. Carrier Gas of 99.999% pure helium. Initial temperature of 80 °C. Temperature ramp of (1) 10 °C/min until 120 °C, and then held for 4 min. Temperature ramp (2) of 20 °C/min until 140 °C, and then held 5 min. Temperature ramp (3) of 2 °C/min until 200 °C, and then held for 45 min. The temperature of the detector was set to 240 °C. The total time of the analysis was 53 min. The detection system was a mass detector.

#### 2.3.4. Identification and Quantification of Fatty Acids

The identification and quantification of fatty acids were developed using a gaseous chromatography technique with Agilent Technologies 7890B G.C. coupled with 5977ª GC/MSD equipment. The proportion was determined with the reference pattern (FAME MIX C4-C24, Supelco, Bellefonte, PA, USA) through a comparison with the NIST14. L library and using a calibration factor.

### 2.4. Statistical Analysis

The data were processed and analyzed with the GraphPad Prism 5.0 program (Graph-Pad Software, San Diego, CA, USA). Analysis of variance was performed using the one-way or two-way ANOVA test, and, when this was significant, the means were compared using Tukey’s test.

## 3. Results

The visual appearance of the different sausages evaluated in this study are shown in Figure 1. The sausages showed differences in appearance, color, and shape characteristic of the areas where they were collected. Visually, the artisanal sausages were no different from the commercial products found on the market.

### 3.1. Proximal Composition and Nutritional Estimations

The proximal composition of the different sausages is shown in Table 1. As a result of the composition and formulation, sausages are considered a high-intermediate moisture food, and the moisture values for this type of product could be between 50–75% [1,31]; all of the sausages evaluated were in this range, oscillating between 62.50 to 70 g/100 g of sample. Also, it is essential to note that all of them had a high water content, likely attributable to the inclusion of starch, fibers, or some flour [32]. The difference in moisture percentage in the samples was possibly due to the amount of water added to each sample according to the formulation of each factory.

The ash content showed differences in all of the sausages, but in a very narrow range (*p* < 0.05) (Table 1); values ranged between 2.54 and 4.31 g/100 g of sample; the ash content in this study was similar to that reported by dos Santos Alves et al. [33] (~3–4%) in a Bologna low-fat sausage formulated with a gel pork skin emulsion, banana flour, and water (ratio of 1:2:2). The differences in ash content could be attributed to the amount of flour usually used in the production of this type of sausages and the mineral of the meat that was used to produce the artisan sausages. Ecuadorian artisan sausages were proven to be a richer source of proteins, and the results showed significant differences (*p* < 0.05). It is noticeable that in these artisan sausages, the protein content was similar to what was presented by Choe and Kim [34] (12.60%) in sausages in which a wheat fiber mixture as a fat replacer was added, and Alvarado-Ramírez et al. [35] who used carrot powder as an ingredient.

The fat content showed differences (*p* < 0.05) ranging from 6.61 to 12.33 g/100 g of sample. In this type of product, the variability in fat content is attributed to the meat and pork back fat used in the formulations; however, the producer decides what kind of sausages it needs to produce, and the amount of total fat content, which should be at most 28% [36]; in this sense, all of the samples were within the range. It was not expected that the sausages in this study would have a lower fat content than the sausages reported by Salazar, Arancibia, Calderón, López-Caballero, and Montero [1] the control sausage of their study; also, our values were lower than those in the studies in which fat was reduced [4]. The content of carbohydrates showed variability (*p* < 0.05); the values in samples varied from 6.65 to 14.16 g/100 g of sample; the values observed could be attributable to the use of flours, starches, and other meat extenders to improve the texture and sensory quality of products [37]. Concerning the caloric content of sausages (Table 1), it was observed that according to the components evaluated, the caloric content was comparable to the values reported by Choi et al. [38] in sausages with makgeolli lees fiber, or by Salcedo-Sandoval et al. [39] in sausages with olive oil, flax, or konjac gels. The World Health Organization [40] recommends that the energy content be composed of a variable contribution of carbohydrates, fat, and protein. In products such as sausages, the proportion is very far from the desirable energy balance; however, it is essential to evaluate which fat is detrimental to human health and to establish a lipid profile to know the natural effect of fat on the nutrition of customers.

### 3.2. Acidity and pH

Acidity variations between sausages were minimal (*p* < 0.01) (Table 1). According to the percentage of lactic acid, the values were between 0.31% to 0.70%. The difference in acidity could be due to the glycogen content in the meat before elaborating the sausages, as lactic acid is produced from glycogen, according to Wang et al. [41]. Lactic acid produced by the fermentation of carbohydrates generates a decrease in pH, inhibiting the growth of harmful microorganisms, precipitating dehydration, slowing water retention, and affecting the color of the finished products. Similar results were reported by Salazar, Arancibia, Calderón, López-Caballero, and Montero [1] in sausages enriched with green banana flour. However, the relationship between lactic acid and pH has not been determined because pH does not only rely on lactic acid, as in cured and refrigerated products. It has been observed that the velocity of the reduction in residual nitrite is related exponentially to the pH and temperature. The reduction rate doubles with each 12 °C temperature increment and each reduction of 0.86 pH units [42,43]. SFAM06 showed a low concentration of nitrites and low pH value; according to the pH in the samples, it was possible to observe that these values were close to neutral. The values obtained in this study were lower than those of the sausages in rural markets in the center of Mexico [44].

### 3.3. Water Activity, Chlorides, and Nitrites

The sausage water activity did not show significant differences (*p* > 0.05) (Table 2). However, although there were no differences, it is essential to note that samples with a high-water activity had a shorter shelf life. The water activity values were typical for these types of gel-emulsion cooked meat products [45]. The growth of *Salmonella* spp was inhibited in water activity values below 0.94, *L. monocytogenes* ≤ 0.94, and *Staphylococcus aureus* < 0.83. The reduction in water activity in meat derivatives depended on the temperature, relative moisture, and, mainly, the formulation [46,47,48]. On the other hand, chlorides ranged between 1.81% to 3.41% (*p* < 0.05). Sample SFAM06 showed the highest value; also, this value was outside of the average allowed by the Ecuadorian Institute for Standardization [36], for which the values ranged between 1.5% to 3.0%; sample SFAM06 was unique and did not comply with the rules; this result was probably attributable to the different quantities of salt added by the sausage producers.

The concentration of nitrites in artisan sausages showed different values (*p* < 0.05); the results showed that the addition of nitrites was likely a problem because the producers were not dosing correctly, even though the values were below those allowed by the Food And Drug Administration [49] and the Codex Alimentarius. Samples SFAM06 and SFPL10 showed the lowest concentration and SFAM05 showed the highest, given the different amounts of nitrites added to the sausage according to each formula. Adding nitrites also produced nitrosomyoglobin, a characteristic red pigment generated when this salt decomposes [50]. The evaluated sausages presented as characteristic red sausages; however, the characteristic red color could not be evaluated because industries use artificial coloring to generate homogeneity in their products.

### 3.4. Color Properties

The visual appearance of the sausages obtained from different factories showed a homogeneous aspect, with differences in color found in some samples (Table 3). The color of meat derivatives was due to biochemical reactions between myoglobin, hemoglobin, and oxygen, as well as the action of nitrites and nitrates [51]. Lightness (L*) values showed significant differences (*p* < 0.05) between the analyzed samples. The sausages showed values of ~50, which means they could not be classified as light or dark. The lightness, as well as other color properties, were difficult to compare with other sausages because of the influence of the formulation, the use of different colorants, and the process. In this sense, the data obtained showed specific information about artisan sausages produced in Tungurahua, Ecuador.

The samples were grouped in the same quadrant of CIE*L*a*b* space based on their positive a* and b* values. The a* values (*p* < 0.05) showed a tendency towards the red color, probably attributable to the application of natural or artificial colorants [49], which are used to obtain homogeneous products and satisfy customers [14,33]. Regarding the b* values, the SFPL07 sample showed the most pronounced values (*p* < 0.05).

Samples SFAM05 and SFPL09 showed higher chroma (C*) values (*p* < 0.05). That is, these sausages had more saturated or intense colors compared with the rest of the samples. On the other hand, samples SFAM03 and SFPM04 showed lower chroma (C*) values (*p* < 0.05). The presence of the meat proteins (beef and pork) used for emulsification and the addition of permitted additives (such as salt replacers) considerably influenced this parameter [52]. Typically, this characteristic is adjusted with the use of coloring according to consumer preferences. For example, sausages with more intense colors in some developing countries are considered better quality [53,54], while in developed countries, the trend is completely the opposite [55].

The hue tone (*h) showed a similar tendency to the chroma (*C). Samples SFAM05 and SFPL09 presented lower h* values (*p* < 0.05) because they were closer to the intense reds in the CIE*L*a*b* space. The rest of the samples show higher values (*p* < 0.05), which indicates they were in the pink zone (around h* = 45°). Also, the different binders used in the formulation played an important role in high C* and h* values, according to a study by Jin et al. [56].

The different coloration in the samples could be due to the variation in the color of the mass formed by the emulsified muscle protein [57]. As for the results obtained by other authors, variable effects have been found in chroma when fat is replaced with another ingredient in meat products [58].

### 3.5. Microbiological Analysis

The counts of mesophilic aerobic microorganisms were 2.5–3.8 log CFU/g for all of the lots. Similar results were found by Salazar, Arancibia, Calderón, López-Caballero, and Montero [1] in sausages with green banana flours, which reported 3 log CFU/g, and Ranucci et al. [59] in sausages with pork enriched with emmer wheat, hazelnut, and almond, which reported 3.05 log CFU/g. These were studies on cooked sausages; however, the formulations were so different that it took effort to compare the results. Despite this, similar results were observed in other meat products; Vienna-homogenized sausages manufactured with different commercial functional additives reached 3.35 and 3.61 log CFU/g for aerobic bacteria when AFX and FPRX commercial additives at minimum concentrations were used [60], and bologna-type fat-reduced turkey sausage with potato starch and pea fiber did not exceed 4 log CFU/g (depending of the formulation) [61]. On the other hand, Cerón-Guevara et al. [62] reported counts that ranged from 4.52 to 6.12 log CFU/g in low-fat Frankfurter-type sausages with mushroom flour (2.5 and 5%); the authors attributed the results to the spore-forming bacteria present in agricultural products.

Mold and yeast counts were evaluated due to the possible inclusion of flour in the formulations producers used for meat extenders, which is common practice in these types of artisan sausages. No microorganisms were found the artisan sausages; however, it is essential to note that the product quality could be affected by the inclusion of flour, as well as its starch content, due to the presence and increment of mold and yeasts, given that these microorganisms use starch as a growth substrate [63]. Macedo et al. [64] reported counts of molds and yeast between 2.78 to 4.41 log CFU/g in hot dog sausages; these results were attributable to the pH, high water activity, and nutrient composition [65]. Likewise, Nkekesi et al. [66] reported fungal counts ranging from 0.0 CFU/g to 9.83 × 10^3^  CFU/g for street-vended grilled beef sausages. The presence of molds and yeasts was usually related to food contamination and the production of mycotoxins, which, in most cases, were not easy to eliminate during processing.

In *Enterobacteria* and *Staphylococcus aureus*, no presence was detected, probably as a result of the heat treatment (85 °C during 30 min) and the addition of nitrites into the formulation, because they inhibited the growth of these microorganisms due to a halt in active transport, oxygen absorption, and oxidative phosphorylation in bacteria such as *S. aureus*, *Escherichia coli*, and *Clostridium botulinum* [67].

### 3.6. Sensory Analysis

Sensory properties are an important parameter for all types of foods; however, when products receive the qualification of being artisan, these parameters become dependent on the point of view of the consumers who purchase the product. Many consumers and producers are not willing to make concessions regarding the aroma, flavor, texture, and juiciness of meat products, especially sausages. The sensory attributes of sausages of 10 factories are shown in Figure 2. The sausages registered an acceptability average score of ~5, indicating that consumers liked the sausages; in this sense, the acceptability results could be attributed to the formulation used by each factory in order to be accepted in the market. It was assumed that each factory had developed a sensory evaluation that allowed them to create the most appropriate formulation that was accepted by consumers. It is important to remember that the objective of this work was to evaluate the quality of these types of artisanal sausages and that they were already for sale on the market. The sensory evaluation allowed us to establish that these products were accepted in the market. No strange odors or flavors were detected in the sausages during the evaluation. The scores tended to be the same as acceptability; that is to say, the judges showed their perception that they liked the aroma of the sausages. These results were attributed to the fact that these types of sausages used similar condiments in their preparation. A literature revision was developed to compare the sausages produced using nutritional components such as fibers, vegetable flour, fruit flour, and other types of flour. The results in different studies showed that when a non-traditional ingredient was added, the sensorial properties tended to decrease. Muchekeza et al. [68] reported that sausages made with quinoa as a binder obtained acceptability values that ranged from 6.71 to 7.14 on a nine-point hedonic scale (extremely like); however, Amaranth had a range of 5.4–5.94, which reflected a neither like nor dislike by consumers. Another study of Frankfurter-type sausages with the addition of carrot paste showed that the aroma and flavor decreased the acceptability when compared with the control [69]. Zaini et al. [70] used banana peel powders to study the effect on the technological functionality, sensory quality, and nutritional quality of chicken sausage; with respect to sensorial properties, the results showed that that the addition of 2–6% of banana peel powder significantly reduced the aroma and appearance of the treated sausage. Choi et al. [71] reported that the inclusion of 3–6% brown rice fiber reduced the overall acceptability of the sausage. The artisan sausages evaluated in this study showed a different behavior compared with the sausages with a differentiated nutritional value and were not only called artisanal to influence consumer perception.

### 3.7. Lipidic Content of Sausages

The variation in the composition of the fatty acids in meat products depends fundamentally on the diet of the animals or the overall dietary patterns [72]. In the fatty acids detected in the artisan sausages of this study (Table 4), the highest percentages corresponded to elaidic, palmitic, linolelaidic, and stearic fatty acids. Palmitic and stearic acids correspond to the saturated fatty acid group, while elaidic and linolelaidic acids are trans-fatty acids. Concerning the different places the sausages were collected from, it was observed that sausages from Pelileo (SFPL07, SFPL08, and SFPL09) and one area of Ambato (SFAM03) showed the highest content of elaidic fatty acid; this fatty acid has attracted much attention because it is the principal trans-fatty acid found in hydrogenated vegetable oils, and it is linked to heart disease [73]. It is important to note that in the other sausages from other places, the elaidic content was also high; likewise, the content of palmitic, stearic, and linolelaidic acids were high in contrast with the other fatty acids, which could probably be attributed to the diets animals received, usually consisting of a particular type of hydrogenated vegetable oil. These types of fatty acid are often created by the partial hydrogenation or elaidinization of vegetable oils [74]. The tendency in composition observed in the sausages could probably show that, independent of the place where sausages were produced, the meat and other components used for sausage production were similar throughout Tungurahua province. The results obtained in this study were close to those reported by Franco et al. [75] in traditional sausages from Galicia; the authors reported the highest total level of fatty acids and included oleic (45% of the total fatty acids), palmitic (21%), linoleic (14%), stearic (13%), and palmitoleic acids (2%). The presence of fatty acids such as oleic acid allowed us to infer that, despite assuming that the fat content was detrimental to health, components were also observed, such as the case of oleic acid, which counteracted the intensity of specific inflammatory processes, as it reduced the production of chemotactic mediators of inflammation [76].

Saturated fatty acids, monounsaturated acids, polyunsaturated acids, and trans-fatty acids showed differences (*p* < 0.05) (Table 5). The sum of monounsaturated and polyunsaturated fatty acids determined the total amount of unsaturated fatty acids (UFA). The SFA/UFA index was calculated, as was the relationship between omega-6 and omega-3 (ω-6/ω-3). The atherogenicity index showed values between 3.34 to 6.36, which could be attributed to the meat used to produce the sausages. The results of this study showed that, independent of the place where the sausages were obtained, the significant components were saturated and trans-fatty acids; this indicated that the artisan sausages probably had the same harmful fat components as other commercial meat derivates, as saturated and trans-fatty acids have been linked to several adverse health effects [77]. Epidemiological and clinical evidence has shown that trans fats are essential for developing cardiovascular diseases and are linked to inflammation, diabetes, and cancer.

On the other hand, unsaturated fatty acids help preserve consumer health [78]. The results obtained in this study were different from the sausages collected in butcher shops in the center of Mexico, which contained 37.95% saturated fatty acids, 48.45% monounsaturated acids, and 13.61% of polyunsaturated acids [44]. Meat products supplied 23% of the total fat, 29% of dietary fatty acids, 21% of monounsaturated fatty acids, and 16% of polyunsaturated fatty acids [79].

The relationship between saturated and unsaturated fatty acids is recommended to be above 2.5 because it is an indicator of high nutritional value [80], Slama et al. [81] reported, for example, that pearl millet had a ratio of unsaturated fatty acid to saturated fatty acid equal to 3.39. Lower levels should be increased to optimize the relationship between polyunsaturated and saturated acids for consumption. In this sense, the relationship between saturated and unsaturated fatty acids found in the artisan sausages made in Tungurahua, Ecuador, indicated that the sausages would not harm consumer health.

The studies showed that the level of ω-6 fatty acids and the relationship between ω-6/ω-3 pointed to a higher risk of obesity related to these parameters, and it played a fundamental role in the increased risk of obesity through eicosanoid metabolites [82]. The optimal relationship between ω-6/ω-3 should be between 5:1 and 10:1 in weight; the samples in this study were below these referred values. The atherogenicity index by Ulbricht and Southgate defined the relationship between the fatty acid content and the capacity to elevate the levels of serum cholesterol, formed by lauric, myristic, and palmitic acids, as well as monounsaturated and polyunsaturated fatty acids, which fulfill a protective role by reducing the risk of developing coronary diseases [83].

## 4. Limitations

The findings of this study are limited to the use of physicochemical parameters, microbiological characteristics, and sensory properties that are considered most suitable for establishing the quality of commercial sausages sold. Likewise, the determination of the fat content was supposed to show whether there were elements that allowed for the term artisanal to be associated with a quality somehow implicitly implied in the name given to these sausages. However, the depth of the study did not allow for a comparison with considerations regarding the development of products with better nutritional and technological quality. However, one of the concerns in developing countries is that consumers should not be deceived by terms that can generate an expectation different from the actual product attributes. The determination of the quality and lipid profile of this study aimed to contribute to the nutritional information of sausages that are marketed in Tungurahua, Ecuador.

## 5. Conclusions

This study evaluated the quality and lipid content of artisan sausages from 10 factories. The results of the nutritional quality indicated that the artisan sausages were in the ranges recommended for commercial sausages concerning moisture, fat, protein, and ash. There was no evidence in the results that indicated that the artisan sausages could have any valuable parameters to establish a better nutritional value; on the contrary, the results could indicate that “artisan” is only the name of the product. As indicators of the quality of production, the pH and acidity did not show significant changes, related to microbiological quality, because none of the microbiological indicators evaluated showed the presence of harmful microorganisms. There were no negative observations for taste or odor in the sensorial parameters, while the acceptability was assessed well. The results of the lipid content showed significant differences, and the results showed a high presence of saturated fatty acids (palmitic, stearic, elaidic, and linolelaidic). The results indicated that the same unsaturated fatty acids were present and were the highest in composition, independent of the place they were produced and collected. Also, the relationship showed that unsaturated fatty acids were primarily found in the artisan sausages. In conclusion, the results of the artisanal sausages evaluated here showed that there was no remarkable characteristic denoting that the premise of the artisanal sausages being healthier was reflected in their composition.

## Figures and Tables

**Figure 1 foods-12-04288-f001:**
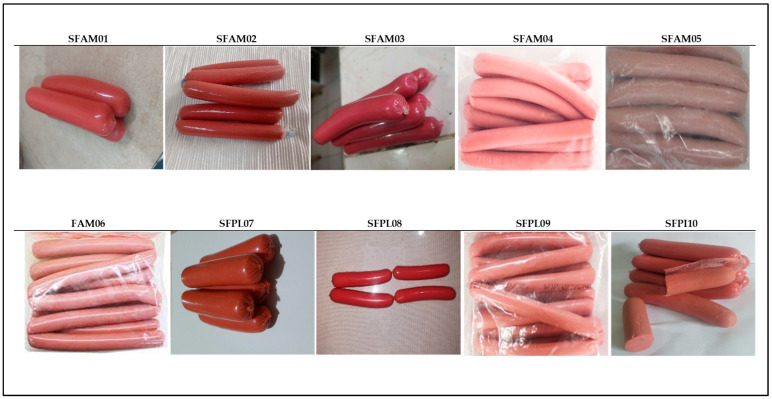
Visual appearance of artisan sausages produced in Tungurahua, Ecuador.

**Figure 2 foods-12-04288-f002:**
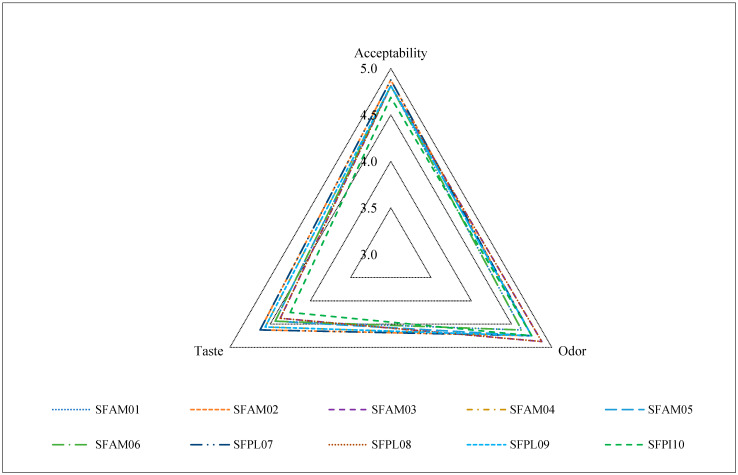
Sensory attributes of commercial sausages obtained from 10 different factories located in Ecuador.

**Table 1 foods-12-04288-t001:** Proximate composition (g/100 g), energy values (Kcal/100 g), and pH and acidity (g/100 g of lactic acid) of artisan sausages produced in Tungurahua, Ecuador.

Parameters	Samples									
	SFAM01	SFAM02	SFAM03	SFAM04	SFAM05	SFAM06	SFPL07	SFPL08	SFPL09	SFPI10
Moisture	66.72 ± 0.01 ^c^	66.28 ± 0.01 ^cd^	66.70 ± 0.32 ^c^	70.00 ± 0.17 ^a^	63.21 ± 0.09 ^f^	67.27 ± 0.01 ^b^	62.50 ± 0.21 ^g^	64.27 ± 0.01 ^e^	65.84 ± 0.14 ^d^	64.13 ± 0.21 ^e^
Protein	9.54 ± 0.05 ^h^	12.97 ± 0.05 ^c^	10.85 ± 0.01 ^e^	10.67 ± 0.02 ^f^	15.70 ± 0.02 ^a^	13.64 ± 0.01 ^b^	10.95 ± 0.09 ^e^	9.66 ± 0.05 ^g^	12.17 ± 0.02 ^d^	12.19 ± 0.02 ^d^
Fat	8.69 ± 0.30 ^ef^	8.01 ± 0.30 ^f^	8.31 ± 0.32 ^f^	6.61 ± 0.28 ^g^	10.18 ± 0.56 ^c^	9.91 ± 0.21 ^cd^	12.33 ± 0.01 ^a^	9.27 ± 0.03 ^de^	9.41 ± 0.16 ^cde^	11.09 ± 0.06 ^b^
Ash	2.68 ± 0.01 ^de^	3.25 ± 0.01 ^c^	3.50 ± 0.10 ^b^	3.45 ± 0.01 ^b^	4.31 ± 0.01 ^a^	2.71 ± 0.05 ^d^	2.73 ± 0.09 ^d^	2.80 ± 0.05 ^d^	2.54 ± 0.04 ^e^	3.37 ± 0.06 ^bc^
Carbohydrates	12.39 ± 0.03 ^b^	9.51 ± 0.03 ^f^	11.10 ± 0.03 ^d^	9.56 ± 0.02 ^f^	6.99 ± 0.02 ^g^	6.65 ± 0.02 ^h^	11.76 ± 0.29 ^c^	14.16 ± 0.13 ^a^	10.26 ± 0.04 ^e^	9.34 ± 0.05 ^f^
Calories	165.93 ± 1.20 ^g^	162.01 ± 0.82 ^h^	162.59 ± 0.35 ^h^	140.42 ± 0.65 ^i^	182.42 ± 0.21 ^c^	170.37 ± 0.38 ^f^	201.84 ± 0.57 ^a^	178.67 ± 0.99 ^d^	174.47 ± 0.71 ^e^	185.89 ± 0.82 ^b^
pH	7.16 ± 0.01 ^a^	7.14 ± 0.01 ^a^	7.10 ± 0.07 ^ab^	6.64 ± 0.47 ^cde^	6.24 ± 0.01 ^e^	6.48 ± 0.01 ^de^	6.65 ± 0.03 ^cde^	6.47 ± 0.01 ^de^	6.97 ± 0.01 ^abc^	6.70 ± 0.01 ^bcd^
Acidity	0.39 ± 0.02 ^e^	0.48 ± 0.01 ^b^	0.42 ± 0.01 ^c^	0.36 ± 0.01 ^e^	0.70 ± 0.01 ^a^	0.42 ± 0.01 ^c^	0.31 ± 0.01 ^f^	0.31 ± 0.01 ^f^	0.47 ± 0.01 ^b^	0.42 ± 0.01 ^c^

The results are the mean ± standard deviation. One-way ANOVA: Different letters (^a.b.^…) in the same row indicate significant differences among samples (*p* ≤ 0.05). Values of proximal composition are presented as g/100 g of sausage.

**Table 2 foods-12-04288-t002:** Results of the physicochemical parameters of the sausage samples.

Sample	Water Activity (a_w_)	Nitrites (ppm)	Chlorides (%)
SFAM01	0.98 ± 0.01 ^a^	100 ± 5.00 ^d^	1.81 ± 0.09 ^d^
SFAM02	0.98 ± 0.01 ^a^	100 ± 8.00 ^d^	2.60 ± 0.12 ^b^
SFAM03	0.98 ± 0.01 ^a^	140 ± 6.00 ^a^	2.62 ± 0.17 ^b^
SFAM04	0.97 ± 0.01 ^a^	130 ± 4.00 ^ab^	2.87 ± 0.20 ^b^
SFAM05	0.96 ± 0.01 ^a^	140 ± 7.00 ^a^	3.41 ± 0.01 ^a^
SFAM06	0.98 ± 0.01 ^a^	110 ± 5.00 ^cd^	1.98 ± 0.09 ^cd^
SFPL07	0.98 ± 0.01 ^a^	122 ± 7.00 ^bc^	2.16 ± 0.02 ^d^
SFPL08	0.98 ± 0.01 ^a^	122 ± 6.00 ^bc^	2.10 ± 0.02 ^cd^
SFPL09	0.98 ± 0.01 ^a^	116 ± 5.00 ^bcd^	1.81 ± 0.06 ^d^
SFPI10	0.98 ± 0.01 ^a^	121 ± 4.00 ^bc^	2.14 ± 0.01 ^c^

The results are the mean ± standard deviation. One-way ANOVA: Different letters (^a.b^....) in the same column indicate significant differences among samples (*p* ≤ 0.05).

**Table 3 foods-12-04288-t003:** Color parameters of artisan sausages from Tungurahua, Ecuador.

Sample	L*	a*	b*	C*	h*
SFAM01	50.09 ± 0.52 ^de^	21.45 ± 0.16 ^c^	16.97 ± 0.16 ^bc^	27.35 ± 0.53 ^bc^	38.35 ± 0.17 ^de^
SFAM02	46.45 ± 0.33 ^g^	23.08 ± 0.08 ^b^	16.52 ± 0.18 ^c^	28.38 ± 0.34 ^b^	35.59 ± 0.09 ^f^
SFAM03	52.20 ± 0.48 ^bc^	16.15 ± 0.27 ^e^	14.82 ± 0.27 ^d^	21.92 ± 0.49 ^e^	42.54 ± 0.28 ^b^
SFAM04	53.98 ± 0.84 ^a^	14.65 ± 0.79 ^f^	13.23 ± 0.77 ^f^	19.74 ± 0.85 ^f^	42.08 ± 0.80 ^b^
SFAM05	49.45 ± 0.51 ^e^	25.91 ± 0.38 ^a^	17.98 ± 0.17 ^b^	31.54 ± 0.52 ^a^	34.76 ± 0.39 ^fg^
SFAM06	47.56 ± 0.44 ^fg^	20.48 ± 0.13 ^cd^	16.27 ± 0.33 ^c^	26.16 ± 0.45 ^cd^	38.46 ± 0.14 ^cd^
SFPL07	51.64 ± 0.14 ^bc^	12.01 ± 0.17 ^g^	22.92 ± 0.48 ^a^	25.88 ± 0.15 ^d^	62.35 ± 0.18 ^a^
SFPL08	51.28 ± 0.47 ^cd^	20.32 ± 0.48 ^d^	16.73 ± 0.37 ^bc^	26.32 ± 0.48 ^cd^	39.47 ± 0.49 ^c^
SFPL09	48.85 ± 0.56 ^ef^	25.54 ± 0.21 ^a^	17.05 ± 0.97 ^bc^	30.71 ± 0.57 ^a^	33.73 ± 0.22 ^g^
SFPI10	52.70 ± 0.29 ^ab^	21.51 ± 0.46 ^c^	16.41 ± 0.42 ^c^	27.05 ± 0.30 ^bcd^	37.34 ± 0.47 ^e^

The results are the mean ± standard deviation. One-way ANOVA: Different letters (^a.b.^…) in the same column indicate significant differences among samples (*p* ≤ 0.05). L* (lightness); a* (red/green chromaticity); b* (yellow/blue chromaticity); C* (chroma); h* (Hue tone).

**Table 4 foods-12-04288-t004:** Fatty acid contents (g/100 g) in the fat of the artisan sausages produced in Ecuador.

Fatty Acids	Artisan Sausages									
(g/100 g)	SFAM01	SFAM02	SFAM03	SFAM04	SFAM05	SFAM06	SFPL07	SFPL08	SFPL09	SFPI10
Lauric	0.21 ± 0.04 ^b^	0.15 ± 0.04 ^b^	0.22 ± 0.09 ^b^	0.12 ± 0.02 ^b^	0.12 ± 0.06 ^b^	0.14 ± 0.00 ^b^	0.11 ± 0.02 ^b^	0.30 ± 0.15 ^a^	0.16 ± 0.05 ^b^	0.10 ± 0.05 ^b^
Myristic	1.34 ± 0.07 ^bcd^	1.69 ± 0.07 ^b^	1.05 ± 0.32	1.25 ± 0.16 ^cd^	2.13 ± 0.01 ^a^	1.68 ± 0.05 ^b^	0.74 ± 0.05 ^e^	1.37 ± 0.24 ^bc^	1.54 ± 0.04 ^bc^	1.30 ± 0.18 ^bc^
Myristoleic	0.31 ± 0.03 ^b^	0.42 ± 0.01 ^a^	-	0.43 ± 0.00 ^a^	0.41 ± 0.03 ^a^	0.21 ± 0.02 ^cd^	0.16 ± 0.01 ^d^	0.21 ± 0.10 ^bc^	0.40 ± 0.08 ^a^	0.15 ± 0.04 ^d^
Pentadecyl	0.36 ± 0.04 ^a^	0.34 ± 0.01 ^abc^	-	0.37 ± 0.05 ^ab^	0.31 ± 0.01 ^abcd^	0.21 ± 0.04 ^d^	0.12 ± 0.02 ^e^	0.19 ± 0.11 ^cd^	0.25 ± 0.05 ^bcd^	0.12 ± 0.00 ^e^
Palmitic	23.21 ± 0.95 ^a^	25.08 ± 0.06 ^a^	22.04 ± 2.40 ^a^	22.74 ± 2.55 ^a^	23.65 ± 0.08 ^a^	24.28 ± 0.75 ^a^	25.22 ± 0.60 ^a^	22.85 ± 2.98 ^a^	18.60 ± 5.59 ^a^	24.46 ± 1.79 ^a^
Palmitoleic	3.65 ± 0.01 ^cd^	4.65 ± 0.21 ^ab^	4.26 ± 0.55 ^bc^	4.93 ± 0.07 ^a^	3.16 ± 0.03 ^de^	3.04 ± 0.16 ^de^	4.91 ± 0.09 ^a^	2.91 ± 0.54 ^de^	2.90 ± 0.28 ^de^	2.36 ± 0.39 ^e^
Margaric	0.72 ± 0.01 ^a^	0.73 ± 0.01 ^a^	-	0.76 ± 0.11 ^a^	0.76 ± 0.00 ^a^	0.58 ± 0.11 ^ab^	0.25 ± 0.02 ^c^	0.32 ± 0.00 ^c^	0.65 ± 0.20 ^a^	0.45 ± 0.06 ^b^
Cis-10-heptadecenoic	0.28 ± 0.03 ^de^	0.35 ± 0.01 ^cd^	-	0.28 ± 0.07 ^de^	0.38 ± 0.03 ^ab^	0.28 ± 0.03 ^de^	-	0.18 ± 0.04 ^e^	0.43 ± 0.04 ^a^	0.32 ± 0.08 ^abc^
Stearic	12.83 ± 0.39 ^bc^	13.17 ± 0.10 ^b^	6.06 ± 1.19 ^d^	13.92 ± 1.35 ^b^	16.07 ± 0.13 ^b^	13.87 ± 0.14 ^b^	7.13 ± 0.86 ^d^	7.84 ± 2.47 ^cd^	12.43 ± 3.33 ^b^	17.60 ± 3.94 ^a^
Oleic	2.22 ± 0.06 ^a^	1.30 ± 0.10 ^c^	0.11 ± 0.16 ^f^	1.44 ± 0.03 ^c^	1.66 ± 0.08 ^b^	0.87 ± 0.16 ^d^	0.30 ± 0.14 ^e^	0.55 ± 0.02 ^e^	2.37 ± 0.01 ^a^	0.32 ± 0.07 ^e^
Elaidic	35.81 ± 1.32 ^c^	38.96 ± 0.061 ^bc^	44.26 ± 3.84 ^ab^	39.18 ± 2.46 ^bc^	39.52 ± 0.36 ^bc^	40.13 ± 1.60 ^bc^	43.87 ± 0.78 ^ab^	43.06 ± 1.11 ^b^	45.15 ± 6.95 ^a^	38.54 ± 3.13 ^bc^
Linolelaidic	17.50 ± 2.00 ^bc^	11.27 ± 0.39 ^fg^	21.06 ± 0.22 ^a^	13.37 ± 1.52 ^ef^	9.97 ± 0.26 ^g^	12.36 ± 0.03 ^efg^	16.37 ± 1.13 ^cd^	18.70 ± 3.00 ^ab^	13.74 ± 1.66 ^de^	12.47 ± 2.02 ^ef^
Arachidic	0.30 ± 0.10 ^b^	-	-	0.14 ± 0.01 ^b^	0.22 ± 0.08 ^b^	0.39 ± 0.36 ^a^	0.10 ± 0.03 ^b^	0.13 ± 0.00 ^b^	-	0.27 ± 0.05 ^b^
Linolenic	0.88 ± 0.28 ^b^	1.56 ± 0.13 ^a^	0.92 ± 0.17 ^bc^	0.67 ± 0.04 ^cde^	0.63 ± 0.17 ^bcde^	0.63 ± 0.30 ^bcd^	0.46 ± 0.01 ^e^	0.80 ± 0.04 ^bcd^	0.67 ± 0.06 ^bcde^	0.52 ± 0.07 ^de^
Cis-11-eicosenoic	0.32 ± 0.05 ^cd^	0.32 ± 0.06 ^cd^	-	0.37 ± 0.07 ^bcd^	0.49 ± 0.10 ^b^	0.68 ± 0.29 ^a^	0.24 ± 0.02 ^d^	0.31 ± 0.02 ^d^	0.39 ± 0.020 ^bcd^	0.50 ± 0.03 ^bc^
Eicosadiene	-	-	-	-	0.49 ± 0.24 ^ab^	0.65 ± 0.64 ^a^	-	0.27 ± 0.07 ^b^	0.36 ± 0.05 ^b^	0.50 ± 0.02 ^b^

The results are the mean ± standard deviation. One-way ANOVA: Different letters (^a.b.^…) in the same column indicate significant differences among fatty acids (*p* ≤ 0.05).

**Table 5 foods-12-04288-t005:** Results of the percentage of fatty acids, omega relationship, and atherogenicity index.

Samples	Saturated Fatty Acids (%)	Unsaturated Fatty Acids (%)	Trans-fatty Acids (%)	Saturated/ Unsaturated	ω-6/ω-3	Atherogenicity Index (I.A.)
SFAM01	39.01 ± 0.04 ^e^	7.67 ± 0.04 ^c^	53.32 ± 0.05 ^e^	5.08 ± 0.01 ^de^	0.00	3.76 ± 0.03 ^f^
SFAM02	41.17 ± 0.04 ^c^	8.61 ± 0.01 ^a^	50.23 ± 0.06 ^fg^	4.78 ± 0.21 ^de^	0.00	3.72 ± 0.01 ^f^
SFAM03	33.64 ± 0.09 ^f^	7.47 ± 0.05 ^d^	58.89 ± 0.02 ^d^	4.50 ± 0.55 ^e^	0.55 ± 0.02 ^c^	3.34 ± 0.02 ^g^
SFAM04	29.38 ± 0.02 ^h^	5.30 ± 0.05 ^h^	65.32 ± 0.04 ^a^	5.55 ± 0.07 ^cd^	0.00	5.00 ± 0.07 ^c^
SFAM05	39.30 ± 0.06 ^d^	8.14 ± 0.01 ^b^	52.56 ± 0.08 ^e^	4.83 ± 0.03 ^de^	0.00	3.42 ± 0.03 ^g^
SFAM06	43.29 ± 0.05 ^b^	7.23 ± 0.04 ^e^	49.49 ± 0.05 ^g^	5.99 ± 0.16 ^bc^	0.78 ± 0.11 ^b^	4.47 ± 0.03 ^e^
SFPL07	41.15 ± 0.02 ^c^	6.36 ± 0.02 ^g^	52.49 ± 0.06 ^e^	6.47 ± 0.09 ^b^	1.02 ± 0.02 ^a^	4.89 ± 0.02 ^cd^
SFPL08	44.30 ± 0.01 ^a^	4.68 ± 0.11 ^i^	51.02 ± 0.03 ^f^	9.47 ± 0.54 ^a^	0.97 ± 0.02 ^a^	6.36 ± 0.04 ^a^
SFPL09	33.68 ± 0.05 ^f^	6.70 ± 0.05 ^f^	60.25 ± 0.02 ^c^	5.55 ± 0.28 ^cd^	0.00	4.67 ± 0.04 ^de^
SFPI10	33.01 ± 0.05 ^g^	5.22 ± 0.01 ^h^	61.77 ± 0.01 ^b^	6.32 ± 0.39 ^bc^	0.33 ± 0.06 ^d^	5.49 ± 0.08 ^b^

The results are the mean ± standard deviation. One-way ANOVA: Different letters (^a.b^…) in the same column indicate significant differences among the samples (*p* ≤ 0.05).

## Data Availability

Data are contained within the article.

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
