# Peer review of "Evaluation of the Quality and Lipid Content of Artisan Sausages Produced in Tungurahua, Ecuador"

_foods, 2023, doi:10.3390/foods12234288_

Round 1

Reviewer 1 Report

Comments and Suggestions for Authors

I reviewed the manuscript titled “Evaluation of the Quality and Lipid Content of Artisan Sausages Produced in Tungurahua-Ecuador. The manuscript is well-written. However, revisions are needed before consideration.

Introduction

This section is not clear. Research gap should be addressed clearly. More recent literature must be added. Novelty of the approach must be highlighted.

Methodology

Table 1. Geographical location of the factories where artisanal sausages are produced: I don’t think this table is necessary. It shows the location of the industry. Do authors think that the location of the industry has an effect on product evaluation?

Meat may not be the same quality in 10 industries. Authors should have taken from different batches produced in the same industry. Although different industries follow the same methodology, they may perform additional approaches to make the product tasty to overcome the competitors.

Above approaches used in manuscript may bias the findings

Proximate analysis: unit of each parameters must be mentioned in methodology

Color: provide citation

All scientific names must be italics

Microbiological Analysis: provide citation

Methyl esterification of fatty acids and Chromatographic determination: provide citation

Line 214: All collected sausages from different locations were in this range in this study. In this range in this study… please revise it

The use of grammar should be revised throughout the manuscript

Table 2: comparison among the same column? It is wrong. I think authors mean is same rows… among factory samples

What is the impact of salt content on the quality of sausages?

Color: discussion must be improved

3.5 Microbiological analysis: discussion must be improved

All scientific names must be in italics

3.6. Sensory analysis: no discussion with available scientific literature. Authors should revise this carefully

Table 5: what does it mean by in the same line indicate? Is it a row or column?

Table 6. Results of the percentage of fatty acids, omega relationship, and atherogenicity index.: authors should perform statistical analysis among samples

Ref:

Scientific names must be in Italics

All references must be in consistent format  

Comments on the Quality of English Language

Minor languages changes are required

Reviewer 2 Report

Comments and Suggestions for Authors

The manuscript titled "Evaluation of the Quality and Lipid Content of Artisan Sau-2 sages Produced in Tungurahua-Ecuador" presents an interesting results form the cognitive perspective. It's worth to present the data to consumers. However, the scientific merit is very low. The research lead to well known conclusions. I would encourage the authors to think over the aims considering the research gap in the literature and then submit the revised manuscript once again.

My further concerns are as follows:

1. Line 41-42: What do you mean by reduced health components?
2. Lines 100-116: Methods should be described in more detail and/or references for the methods presented.
3. Lines 121-129: There's a methodology for meat color measurement. Why haven't you used it?
4. Lines 278-279: High water activity due to low salt content? A bit controversial. Sodium chloride is used to retain water in sausage.
5. Table 5: How can you present fatty acid profile without using FID detector (Line 198)?

Comments on the Quality of English Language

There are some minor mistakes that should be corrected.

Reviewer 3 Report

Comments and Suggestions for Authors

Line 16: Please reconsider using the abbreviations: AGS and AGI which are not repeated  in the text.

Line 84: The lipid content isn’t the same as fatty acid composition. This remark also applies to the title of the manuscript.

Line 93: The manufacture of the sausages was describe very overall. What kind of meat and fat was used for the production of sausages (pork/beef)? It should be specified in the p. 2. Material and methods.

Line 95 and line 349: The cooking time should be the same (20 or 30 minutes).

Line 96-97: How long these sausages were refrigerated stored before testing? Add this information to the text.

Line 134-137 and p. 3.5 The names: Enterobacteriaceae, St. aureus, Clostridium botulinum, Escherichia coli should be in italic.

Line 141: Add some information about conditions of the sensory assessment (e.g. light, temperature) and the qualifications of the sensory evaluators.

Line 148: It should be explain: “grilled samples”; according to p. 2.1. Materials these sausages were cooked in water bath.

Line 408: Incorrect sentence.

Line 418-419: The sentence is incomplete.

Line 431: “The relation between…. is recommended to be above 2.5” – add some literature.

The conclusions require improvement:

Line 461: The color of the sausages was not assessed in sensory analysis.

Line: 470-472: These sentence is ambiguous.

Line: 467-470.  This conclusion can not be drawn. Artisanal and commercial sausages were not compared in this study.

Round 2

Reviewer 1 Report

Comments and Suggestions for Authors

The quality of the manuscript is now improved after addressing the reviewer suggestions. In my opinion, this version is accepted for possible publication consideration. 

Reviewer 2 Report

Comments and Suggestions for Authors

The authors have improved the manuscript. However, to draw conclusions regarding the artisan products reference material is needed. Therefore, I recommend to do the very same research on conventional products,  compare the results and submit the modified manuscript. 

My further minor concerns are following: 
1. I am aware that the composition of fatty acids can be determined using GC-MS. However, the results should be presented in absolut values, not percentages that suggest fatty acid profile. 

2. I recommend to use g/100 g rather than %  

3. Some sentences need to be modified as they don't make sense (lines 88-89, 98-100)

Comments on the Quality of English Language

There are still some issues that need to be corrected e.g. lines 88-89, 98-100, 105.

Reviewer 3 Report

Comments and Suggestions for Authors

Thank you for your answers. I accept the revised version of the manuscript.
